# Genome-wide association meta-analysis identifies *GP2* gene risk variants for pancreatic cancer

Yingsong Lin (ID) et al.[#]

Pancreatic cancer is the fourth leading cause of cancer-related deaths in Japan. To identify risk loci, we perform a meta-analysis of three genome-wide association studies comprising 2,039 pancreatic cancer patients and 32,592 controls in the Japanese population. Here, we identify 3 (13q12.2, 13q22.1, and 16p12.3) genome-wide significant loci ($P < 5.0 \times 10^{-8}$), of which 16p12.3 has not been reported in the Western population. The lead single nucleotide polymorphism (SNP) at 16p12.3 is rs78193826 (odds ratio = 1.46, 95% confidence interval = 1.29-1.66, $P = 4.28 \times 10^{-9}$), an Asian-specific, nonsynonymous *glycoprotein 2* (*GP2*) gene variant. Associations between selected *GP2* gene variants and pancreatic cancer are replicated in 10,822 additional cases and controls of East Asian origin. Functional analyses using cell lines provide supporting evidence of the effect of rs78193826 on KRAS activity. These findings suggest that *GP2* gene variants are probably associated with pancreatic cancer susceptibility in populations of East Asian ancestry.

[#]A list of authors and their affiliations appears at the end of the paper.

With 35,390 related deaths in 2018, pancreatic cancer is the fourth leading cause of cancer deaths in Japan, after lung, colorectal, and stomach cancers[1]. The incidence and mortality rates of pancreatic cancer have increased steadily over the past decades, while those of other gastrointestinal cancers have shown decreasing trends[1]. Despite the increasing burden levied by pancreatic cancer, few modifiable risk factors other than smoking and type 2 diabetes mellitus (T2D) have been identified, and the 5-year survival proportions remain the worst (<10%) among major malignancies.

Genome-wide association studies (GWASs) have increasingly revealed associations of pancreatic cancer susceptibility with inherited genetic variations. Since the first GWAS, conducted by the PanScan consortium, identified common variants in the gene coding for the ABO blood group system in 2009[2], at least 23 genome-wide significant susceptibility loci have been linked to the risk of pancreatic cancer[3]. However, owing to the smaller sample sizes of the relevant GWASs, fewer loci have been identified for pancreatic cancer than for other common cancers, including breast and colorectal cancers[4,5]. Furthermore, the risk variants identified to date explain approximately 13% of the total heritability on the basis of GWAS-identified single-nucleotide polymorphisms (SNPs) in individuals of European ancestry[6]. These observations suggest that additional risk loci can be identified by increasing the sample size, as evidenced by the trend in the numbers of variants reported by the PanScan and PanC4 consortia. It is also important to expand the GWASs to populations of non-European ancestry because of differences in minor allele frequencies (MAFs) and patterns of linkage disequilibrium (LD) across diverse populations[7]. In fact, previous GWASs focusing exclusively on populations of Eastern Asian ancestry led to the identification of additional susceptibility loci for breast and colorectal cancers[8,9].

The majority of the risk loci for pancreatic cancer were discovered in the PanScan and PanC4 GWASs, which included populations of European ancestry. Only two GWASs have been conducted in East Asian populations: one in China[10] and one in Japan[11]. A total of eight risk loci (five genome-wide significant loci and three loci with suggestive evidence of association) have been identified for pancreatic cancer, but these loci were not replicated in a previous study using samples from European

populations[12]. Therefore, the role of common susceptibility loci in East Asian populations remains uncertain and needs further exploration.

To detect additional susceptibility loci for pancreatic cancer, we perform a meta-analysis combining all published and unpublished GWAS data in Japan, followed by a replication study involving a Japanese population as well as other populations of East Asian origin. We identify 3 (13q12.2, 13q22.1, and 16p12.3) genome-wide significant loci ($P < 5.0 \times 10^{-8}$) and 4 suggestive loci ($P < 1.0 \times 10^{-6}$) for the risk of pancreatic cancer. We replicate the associations between selected $GP2$ gene variants at 16p12.3 in 10,822 additional cases and controls of East Asian origin and further explore the functional impact of the top SNP rs78193826 of the $GP2$ gene. We also demonstrate pleiotropic effects of the $GP2$ variants. Together, these findings indicate that $GP2$ gene variants are probably associated with pancreatic cancer susceptibility in populations of East Asian ancestry.

## Results

**GWAS meta-analysis and replication.** After imputation and quality control of individual subject genotype data, we performed a meta-analysis of 3 Japanese GWASs comprising data from 2039 pancreatic cancer patients and 32,592 controls for 7,914,378 SNPs (Supplementary Table 1 and Supplementary Data 1). Genomic control adjustment was not applied because there was little evidence of genomic inflation (lambda = 1.02, Supplementary Fig. 1). We observed genome-wide significant ($P < 5.0 \times 10^{-8}$) association signals at 3 loci (13q12.2, 13q22.1, and 16p12.3; Fig. 1 and Table 1), for which the genes nearest the lead SNP were $PLUT$ ($PDX1$-$AS1$) and $PDX1$ on 13q12.2, $KLF5$ and $KLF12$ on 13q22.2, and $GP2$ on 16p12.3. In addition, 4 loci (1p13.2 ($WNT2B$), 2p12 ($CTNNA2$), 3p12.3 ($ROBO2$), and 9q34.2 ($ABO$)) showed suggestive evidence of associations ($P < 1.0 \times 10^{-6}$, Supplementary Table 2). Among these risk loci, genome-wide significant associations were observed for 10 SNPs at 16p12.3 (rs78193826, rs117267808, rs73541251, rs4609857, rs4544248, rs4632135, rs4420538, rs73541271, rs4383154, and rs4383153; Supplementary Data 2). The odds ratio (ORs) for these variants ranged from 1.43 to 1.47, indicating generally stronger associations in this region than for GWAS associations for variants identified in previously published GWASs. The lead SNP here is

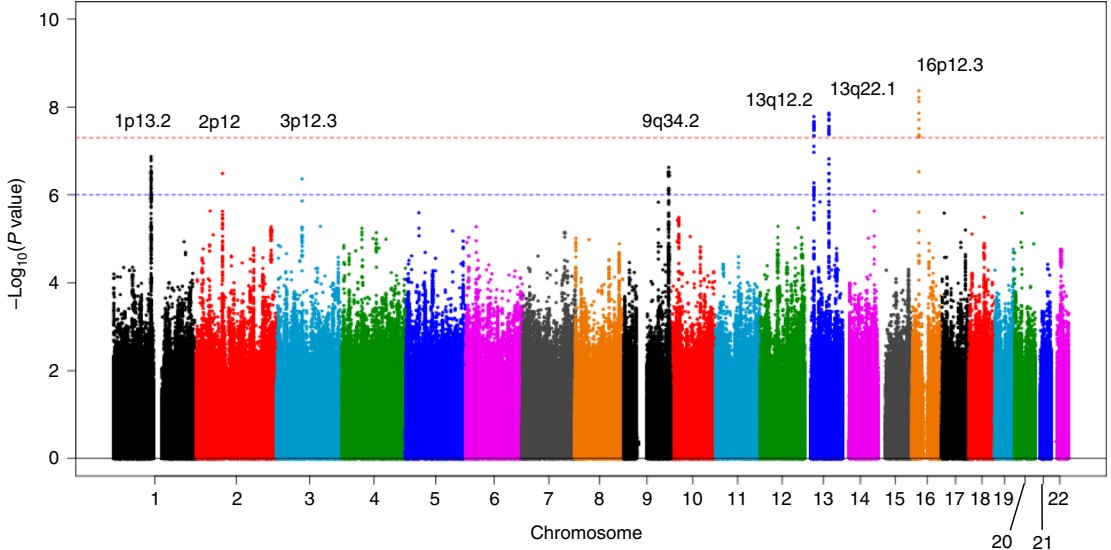

**Fig. 1 Manhattan plot for the meta-analysis.** The horizontal red line represents the genome-wide significance threshold ($\alpha = 5 \times 10^{-8}$). The horizontal blue line represents the suggestive significance threshold ($\alpha = 10^{-6}$).

**Table 1 Genome-wide significant risk loci for pancreatic cancer in the meta-analysis of three Japanese GWASs.**

| SNP | Locus | Chr | Position | Gene | Alleles | | Study | $r^2$ | RAF | | OR (95% CI) | P value | $I^2$ | HetP value |
|---|---|---|---|---|---|---|---|---|---|---|---|---|---|---|
| | | | | | Risk | Non-risk | | | Case | Control | | | | |
| rs147905965 | 13q12.2 | 13 | 28474234 | PLUT | AT | A | JaPAN | 0.987 | 0.237 | 0.194 | 1.30 (1.14–1.47) | $6.09 \times 10^{-5}$ | | |
| | | | | | | | NCC | 0.941 | 0.242 | 0.184 | 1.47 (1.21–1.78) | $1.21 \times 10^{-4}$ | | |
| | | | | | | | BBJ | 0.987 | 0.220 | 0.187 | 1.18 (1.00–1.39) | 0.051 | | |
| | | | | | | | Meta-analysis | | | | 1.29 (1.18–1.41) | $1.66 \times 10^{-8}$ | 28.3 | 0.248 |
| rs9543325 | 13q22.1 | 13 | 73916628 | KLF5, LINC00392 | C | T | JaPAN | 0.999 | 0.555 | 0.495 | 1.28 (1.15–1.42) | $6.01 \times 10^{-6}$ | | |
| | | | | | | | NCC | 1.000 | 0.570 | 0.504 | 1.30 (1.12–1.52) | $6.72 \times 10^{-4}$ | | |
| | | | | | | | BBJ | 0.999 | 0.541 | 0.519 | 1.12 (0.98–1.29) | 0.093 | | |
| | | | | | | | Meta-analysis | | | | 1.24 (1.15–1.33) | $1.38 \times 10^{-8}$ | 24.9 | 0.264 |
| rs78193826 | 16p12.3 | 16 | 20328666 | GP2 | T | C | JaPAN | 0.999 | 0.111 | 0.076 | 1.56 (1.31–1.87) | $8.67 \times 10^{-7}$ | | |
| | | | | | | | NCC | 0.941 | 0.098 | 0.068 | 1.56 (1.17–2.09) | 0.003 | | |
| | | | | | | | BBJ | 1.000 | 0.095 | 0.075 | 1.26 (0.99–1.59) | 0.055 | | |
| | | | | | | | Meta-analysis | | | | 1.46 (1.29–1.66) | $4.28 \times 10^{-9}$ | 14.5 | 0.310 |

OR values represent the increased risk of pancreatic cancer per risk allele copy for each SNP. $r^2$ values indicate quality of imputation metric.
OR odds ratio, CI confidence interval, Chr chromosome, RAF risk allele frequency, HetP value P value from test of heterogeneity.

rs78193826, a nonsynonymous variant of the *GP2* gene (Fig. 2). The risk increased by 46% per copy of the minor T allele (OR = 1.46, 95% confidence interval (CI) = 1.29–1.66, $P = 4.28 \times 10^{-9}$; Table 1). Regional association plots for the other loci are shown in Supplementary Fig. 2. According to the 1000 Genomes Project Phase 3 database, rs78193826 is polymorphic, with the MAF ranging from 3.9% to 6.6% in Asian populations, compared with much lower MAFs (<0.1%) in other human populations (Supplementary Data 2). LD maps of these 10 SNPs at 16p12.3 are shown in Supplementary Fig. 3. Complete LD between nine of these SNPs (all except rs4420538) was observed in the Japanese population. Among the ten SNPs in this region, only rs4383153 had available association summary statistics in the previous PanScan and Panc4 consortia publications[2,13], but this SNP was negligible and not significantly associated with pancreatic cancer risk (Supplementary Table 4). The functional annotation results for the ten SNPs at 16p12.3 are shown in Supplementary Data 2. The lead SNP rs78193826 was classified as "damaging" according to the Sifting Intolerant from Tolerant (SIFT) algorithm and as "possibly damaging" by Polymorphism Phenotyping v2 (Poly-Phen-2). Moreover, the estimated Combined Annotation-Dependent Depletion (CADD) score was 20.3. For replication, we selected four SNPs (rs78193826, rs73541251, rs117267808, rs4632135) that met either of the following criteria: (1) exonic SNP or (2) intronic SNP with a score of ≤3 according to the RegulomeDB database. We sought the replication of associations between these 4 SNPs and pancreatic cancer in populations of East Asian ancestry, including Japanese, Japanese American, and Chinese subjects, using an additional sample of 1926 cases and 8896 controls drawn from 6 independent studies. All 4 SNPs were significantly associated with pancreatic cancer risk ($P < 0.05$) in the combined replication analysis (Supplementary Table 3), with the ORs for the lead SNP rs78193826 shown for each study cohort in Fig. 3.

**Functional characterization of the *GP2*-coding variant**. We explored the functional impact of the identified coding variant rs78193826 in the GP2-expressing pancreatic cancer cell line PaTu 8988s. The single-nucleotide change from G to A at codon 282 of the GP2 gene was induced using clustered regularly interspaced short palindromic repeats (CRISPR)-Cas9-mediated homologous recombination, which enabled the generation of genome-edited PaTu 8988s cells (GP2_V282M PaTu 8988s, Fig. 4a). With RNA-sequencing (RNA-seq) analysis of two separate clones established from wild-type (GP2_WT, serving as a control group) and genome-edited (GP2_V282M) PaTu 8988s cells, we demonstrated consistently different gene expression patterns between GP2_WT and GP2_V282M cells (Fig. 4b, Supplementary Data 3). Gene set enrichment analysis (GSEA) using the collections of hallmark (H) and oncogenic gene sets (C6) in MSigDB v6.2[14] yielded numerous significantly enriched gene sets for GP2_WT cells. Among them, gene sets related to *KRAS*-activating mutations appeared in both the H and C6 collections (Supplementary Data 4 and 5). Because *KRAS* mutation is the most frequently observed mutation and a critical initiating event in pancreatic cancer[15], we focused on these gene sets. Genes in HALLMARK_KRAS_SIGNALING_DN and KRAS.50_UP. V1_DN were significantly expressed at low levels in GP2_V282M PaTu 8988s cells (Fig. 4c). We recapitulated the downregulation of three genes (*KLK7*, *BMPR1B*, and *KLK8*) in HALLMARK_K-RAS_SIGNALLING_DN (Fig. 4d) in three independent GP2_V282M clones compared to GP2_WT clones by quantitative real-time (qRT)-PCR, suggesting that the monoclonal findings were not due to selection bias (Fig. 4e). We also excluded the possibility of vector transfection–induced off-target effects or

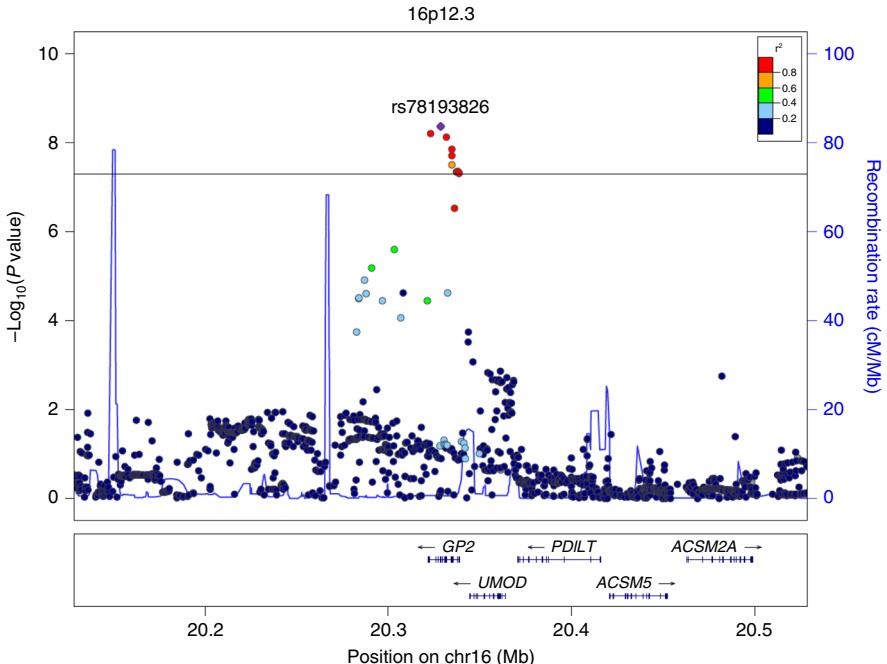

**Fig. 2 Regional association plot for the 16p12.3 locus identified in the meta-analysis.** The vertical axis indicates the −log$_{10}$(P value) for the assessment of the association of each SNP with pancreatic cancer risk. The colors indicate the LD (r$^2$) between each lead SNP and neighboring SNPs based on the JPT population in the 1000 Genomes Project Phase 3. LD linkage disequilibrium, SNP single-nucleotide polymorphism, JPT Japanese.

| Study | N(case) | N(control) | OR (95% CI) |
|---|---|---|---|
| GWAS | | | |
| JaPAN | 943 | 3057 | 1.56 (1.31–1.87) |
| NCC | 674 | 674 | 1.56 (1.17–2.09) |
| BBJ | 422 | 28,861 | 1.26 (0.99–1.59) |
| Meta-analysis (GWAS) | 2039 | 32,592 | 1.46 (1.29–1.66) |
| Replication | | | |
| JaPAN | 507 | 879 | 1.26 (0.94–1.68) |
| JEPA vs HERPACC | 299 | 934 | 1.35 (0.98–1.86) |
| JMICC | 82 | 249 | 2.11 (1.15–3.84) |
| JPHC | 85 | 2493 | 0.87 (0.46–1.65) |
| Shanghai | 770 | 744 | 1.43 (1.11–1.84) |
| MEC | 183 | 3597 | 0.85 (0.56–1.31) |
| Meta-analysis (replication) | 1926 | 8896 | 1.29 (1.11–1.49) |

**Fig. 3 Forest plot of the association of rs78193826 with pancreatic cancer risk.** Study-specific estimates and summary associations are shown. The vertical gray line indicates the null or OR = 1 effect. Horizontal lines through the rectangles indicate the 95% confidence interval. Diamonds represent the overall effect size for each meta-analysis. The width of the diamond spans the 95% confidence interval.

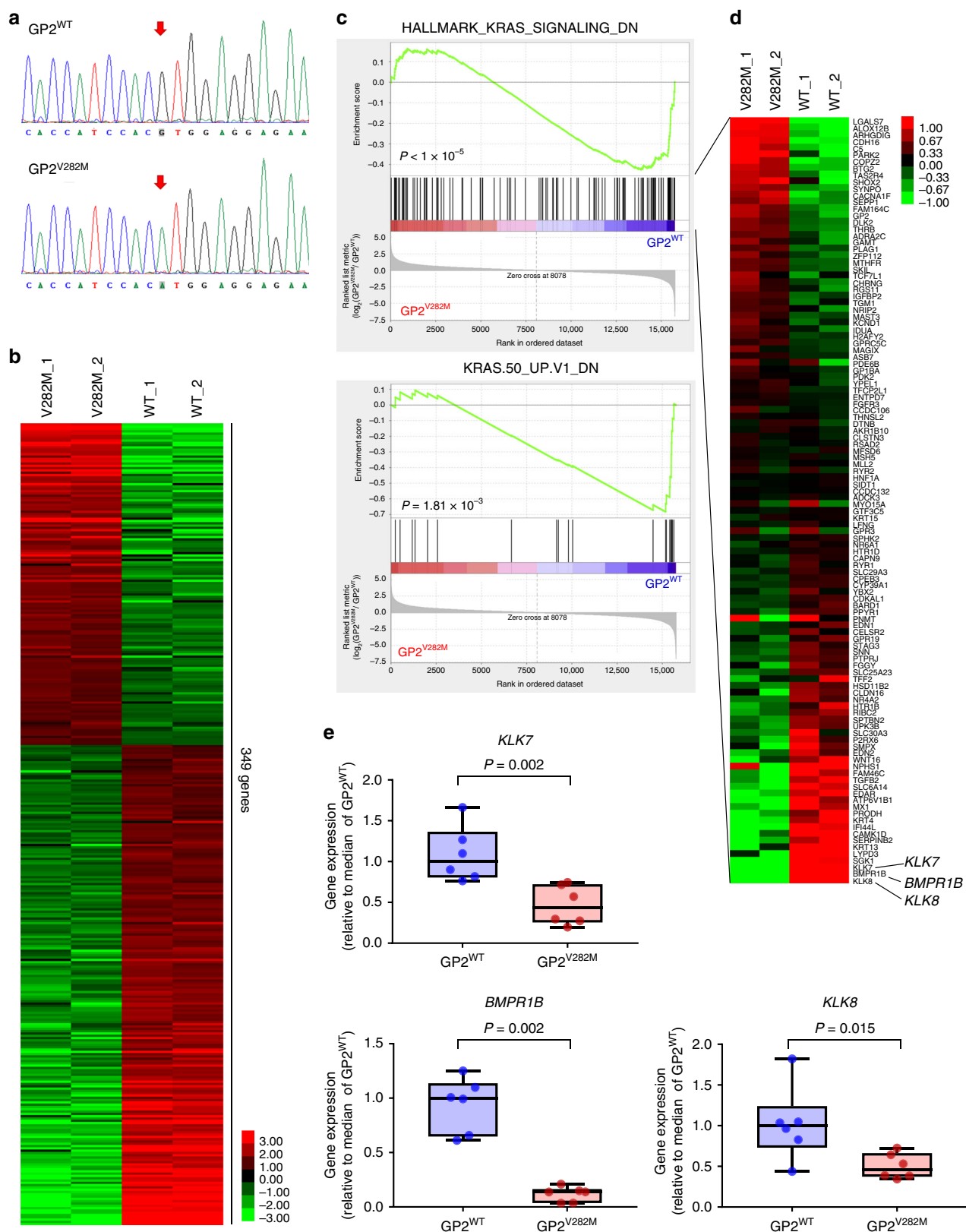

non-specific effects on the expression of genes in specific pathways (Supplementary Figs. 9 and 10, Supplementary Note 1).

**Pleiotropic effects of *GP2* variants.** Epidemiological studies have consistently shown that longstanding T2D is associated with a mildly increased risk of pancreatic cancer[16]. Recently, 88 genetic variants were reported in a GWAS meta-analysis of T2D in the Japanese population[17]. We found that the top 3 SNPs at 16p12.3 (rs78193826, rs117267808, and rs73541251) were also genome wide significantly associated with the risk of T2D in the latest GWAS comprising 191,764 Japanese subjects (Supplementary

**Fig. 4 Functional characterizations of rs78193826. a** Sanger chromatogram showing the WT sequence (top) and the nucleotide mutation from G to A at valine-282 (bottom) of the *GP2* gene. **b** Heatmap depicting the expression of the significantly differentially expressed genes (empirical FDR < 0.10) in GP2_V282M cells compared to the GP2_WT cells ($n = 2$: biological replicates for each group). The expression level was converted to the $\log_2$(RPKM) value, and the mean value of the $\log_2$(RPKM) for each gene across four cells was subtracted. **c** Scatterplot depicting the GSEA for enrichment of *KRAS* signaling-associated gene sets differentially expressed between GP2_WT and GP2_V282M Patu-8988s cells. Two *KRAS*-related signatures are shown. Black vertical lines indicate gene hits, and the ranked list metric based on log2fdc from GFOLD is depicted as a gray line plot. *P* values were calculated by permutation of genes. **d** Heatmap depicting the expression of the genes in the HALLMARK_KRAS_SIGNALING_DN signature. The expression level was converted to the log2(RPKM) value, and the mean value of the log2(RPKM) for each gene across four cells was subtracted. Three genes differentially expressed as log2 fold-change >3 are indicated. **e** qRT-PCR validation of the three genes (**d**) using independent clones of GP2_WT and GP2_V282M. Each group is comprised of $n = 6$ independent samples ($n = 3$ biologically independent cells (clones) with biological duplicates for each clone; each sample was analyzed in a technically duplicate manner). Data are displayed as box and whiskers plots: the box extends from the 25th to 75th percentiles, the middle line represents the median, and the whiskers extend from the minimum to the maximum value. *P* values were determined from exact Wilcoxon rank-sum tests (two sided). Data shown are representative of two independent experiments with similar results. WT wild type, FDR false discovery rate, PPKM, reads per kilobase of exon per million mapped reads; GFOLD, generalized fold change for ranking differentially expressed genes from RNA-seq data, qRT-PCR quantitative real-time reverse transcription-polymerase chain reaction, SE standard error. Source data are provided as a Source data file.

Table 5). In addition, these 3 SNPs were significantly associated with hemoglobin A1c (HbA1c; $P < 1 \times 10^{-4}$) and blood glucose levels ($P < 0.01$), which were included in other GWASs of quantitative traits in 42,790 and 93,146 Japanese subjects, respectively[18]. Among 82 T2D-related SNPs, rs117267808 at 16p12.3 (*GP2*) and rs2290203 at 15q26.1 (*PRC1-AS1*) were significant after Bonferroni correction in our GWAS meta-analysis ($P < 0.0006$, Supplementary Data 6). Among 15 blood glucose-related SNPs, rs4581570 at 13q12.2 (*PDX1*) was significant after Bonferroni correction ($P < 0.0033$, Supplementary Data 7). In addition, none of the 25 HbA1c-associated (Supplementary Data 8) and 76 body mass index (BMI)-associated SNPs (Supplementary Data 9) were significant in our GWAS meta-analysis.

**Mendelian randomization (MR) analysis**. To examine whether the associations of metabolic traits with pancreatic cancer are consistent with a systematic association, we performed a MR analysis with the inverse variance-weighted (IVW) and MR–Egger methods. Inconsistent results were observed for T2D, with no significant association between the SNP index T2D and pancreatic cancer risk based on the IVW method (Fig. 5a). Although the MR–Egger analysis yielded a significant association, the intercept differed significantly from zero ($P < 0.05$, Supplementary Fig. 4a), suggesting the presence of horizontal pleiotropy. Applying the MR Pleiotropy RESidual Sum and Outlier (MR-PRESSO) test enabled the detection of two outlying SNPs (rs117267808 at 16p12.3 (*GP2*) and rs2290203 at 15q26.1 (*PRC1-AS1*); Supplementary Data 6). After correction for horizontal pleiotropy via outlier removal, the association remained nonsignificant in the IVW analysis ($\beta \pm SE = 0.08 \pm 0.06$, $P = 0.16$). In contrast, the HbA1c genetic index level appeared to be related to an increased risk of pancreatic cancer, on the basis of significant results with both the IVW and MR–Egger methods (Fig. 5b and Supplementary Fig. 4b). In addition, we found no significant associations between genetic index levels for the other two metabolic factors (blood glucose and BMI) and pancreatic cancer risk (Supplementary Fig. 5).

**Gene-based GWAS**. To complement the SNP-based GWAS, we performed a gene-based GWAS using MAGMA[19] (Supplementary Fig. 6). Among 17,581 genes, we confirmed the significant associations for *GP2* and *WNT2B* identified by the SNP-based GWAS. In addition, a significant association (Bonferroni-corrected $P = 2.84 \times 10^{-6}$) for the gene *KRT8* was observed (Supplementary Table 6 and Supplementary Figs. 6 and 7), and this association was replicated in the combined PanScan I and PanScan II datasets ($P = 0.024$)[13].

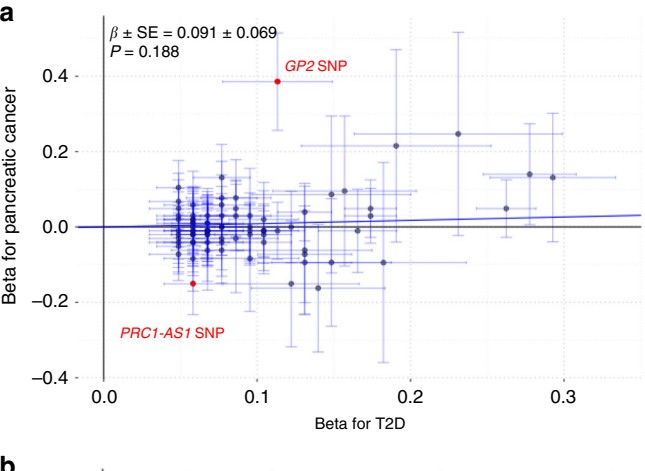

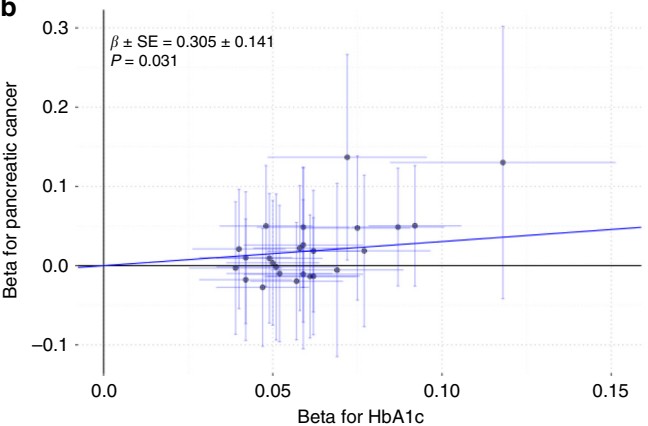

**Fig. 5 Mendelian randomization analysis for the T2D–pancreatic cancer associations. a** The results from 82 type 2 diabetes (T2D)-associated SNPs. Red dots indicate the outlying SNPs detected by MR-PRESSO. Beta values represent the log odds ratio (OR) of T2D or pancreatic cancer per risk allele copy for each SNP. **b** Twenty-five HbA1c-associated SNPs. Beta values represent the change in the rank-based inverse normal transformed values of the HbA1c level per risk allele copy for each SNP. MR Mendelian randomization, T2D type 2 diabetes, MR-PRESSO Mendelian Randomization Pleiotropy RESidual Sum and Outlier, HbA1c hemoglobin A1c.

**Replication of PanScan and PanC4 consortia risk loci**. We also examined the previously published pancreatic cancer risk loci from the PanScan and PanC4 consortia[20], noting that 5 of those 19 SNPs were statistically significant after Bonferroni correction

($P < 0.0026$, rs13303010 at 1p36.33, rs505922 at 9q34.2, rs9581943 at 13q12.2, rs7214041 at 17q24.3, and rs9543325 at 13q22.1) in our GWAS meta-analysis (Supplementary Data 10). Notably, we confirmed the significant association ($P = 3.84 \times 10^{-5}$) between rs505922 of the *ABO* locus and pancreatic cancer risk.

## Discussion

The role of inherited common genetic variations in pancreatic cancer susceptibility remains incompletely understood. We identified and replicated a risk locus at 16p12.3 by combining three GWAS datasets of East Asian populations. Furthermore, we provided evidence that the identification of this locus can be attributed to the observed differences in the MAF of the lead SNP (rs78193826) at 16p12.3 and the LD structure in this region across ethnic populations.

Little overlap has been observed when risk loci reported from previous Chinese or Japanese GWASs are compared with those reported in the PanScan GWASs[2]. By including more than twice the number of case patients than were included in previous Japanese or Chinese GWASs, as well as by using imputed SNP data, we replicated the majority of significant loci discovered for pancreatic cancer in the PanScan and PanC4 consortia GWASs (Supplementary Data 10), including the well-established *ABO* locus. Moreover, for most variants, the direction and magnitude of the associations in our GWAS meta-analysis of Japanese subjects were consistent with those in populations of European ancestry. These findings suggest that GWAS-identified variants at many loci are shared across ancestral groups and that lack of replication may be due to insufficient sample sizes in previous Chinese or Japanese GWASs.

The MAF of rs78193826 in cases varies considerably across populations in the replication cohorts when compared with that in control subjects. The main reason may be due to random variation caused by a small sample size. As shown in Supplementary Table 3, the MAF was higher in controls than in cases in the Japan Public Health Center-based Prospective Study (JPHC), generating an opposite direction in effect size from that observed in other replication cohorts. However, the overall positive association for rs78193826 was replicated in the analysis combining all replication cohorts. Another possible reason is population stratification, but the Multiethnic Cohort Study (MEC) results were adjusted for the principal component. After excluding the JPHC and MEC studies from the replication cohorts, we found no large variations in MAF in the cases.

Several lines of evidence indicate that rs78193826, which lies within the *GP2* gene on 16p12.3, may be associated with pancreatic cancer risk. First, this variant is nonsynonymous; the nucleotide mutation from C to T causes an amino acid change from valine to methionine, which could affect protein structure and function. Second, functional annotations in several databases consistently indicate that this variant is likely pathogenic. Third, the observed differences in the MAF of rs78193826 as well as the LD structure across different ethnic populations provide indirect evidence supporting its role as a significant variant in the Japanese population. The frequency of the minor T allele of rs78193826 is 0.1% in populations of European ancestry but 7% in the Japanese population. Given this apparent difference in the MAF, rs78193826 could not have been identified in the PanScan GWASs, although the PanScan GWASs included a much larger sample size than our GWAS. Of the 10 SNPs in LD in this region, only rs4383153 has association summary statistics available in PanScan publications; however, no significant associations were observed between this SNP and pancreatic cancer risk (Supplementary Table 4). While complete LD between rs4383153 and rs78193826 was evident in the Japanese population, no LD data

were available for these two SNPs in the European ancestry populations (1000 Genomes Project Phase 3 CEU).

Genetic variations in the *GP2* gene have been linked to several phenotypes in addition to pancreatic cancer. The SNP rs12597579, located ~60 kb downstream of *GP2*, has been associated with BMI in a GWAS including East Asians[21]. However, rs12597579 was not in LD with rs78193826 ($r^2 = 0.003$, calculated from Japanese samples in the 1000 Genomes Project Phase 3), suggesting that rs12597579 may have functions different from those of rs78193826. Perhaps coincidentally, the lead variant (rs117267808) in the *GP2* gene identified in the latest GWAS meta-analysis of T2D in the Japanese population was also identified in our GWAS meta-analysis of pancreatic cancer (Supplementary Table 5). Of the 82 T2D-related SNPs, 2 showed significant associations with pancreatic cancer, suggesting that pancreatic cancer and T2D may share specific genetic susceptibility factors.

The identified lead SNP (rs78193826) is in the coding sequence for the GP2 protein, which is present on the inner surface of zymogen granules in pancreatic acinar cells[22]. GP2 is a glycosylated protein of ~90 kDa that contains multiple sites, such as an asparagine-linked glycosylation site, a zona pellucida domain, and a glycosylphosphatidylinositol linkage to the membrane[22]. During the secretory process, GP2 is cleaved from the membrane and secreted into the pancreatic duct along with other digestive enzymes. The expression of GP2 is extremely high in normal pancreatic tissues compared with that in other tissues (Supplementary Fig. 8)[23]. Furthermore, a previous RNA transcriptome analysis revealed that pancreatic tumor tissues have a decreased level of GP2 expression compared with adjacent benign pancreatic tissues[24].

The functional characterization of GWAS-identified SNPs remains a challenge. We conducted a series of experiments to examine the possible functional impact of the nonsynonymous lead SNP rs78193826 on global gene expression. RNA sequencing (RNA-seq) analysis revealed consistently differential gene expression patterns between genome-edited GP2_V282M cells and control GP2_WT cells (Supplementary Data 3). Among many significantly enriched gene sets identified for GP2_WT cells in the GSEA, *KRAS*-related gene sets stood out because of the well-known role of *KRAS* mutation in pancreatic carcinogenesis (Supplementary Data 4 and 5)[15]. Interestingly, gene signatures related to *KRAS*-activating mutations (HALLMARK_KRAS_-SIGNALING_DN and KRAS.50_UP. V1_DN) in two distinct collections (H and C6) showed significant enrichment in GP2_WT cells. Collectively, these experimental findings suggest that the functional relevance of rs78193826 may involve modulation of KRAS activity. Given that a very high frequency (>93%) of *KRAS* mutations has been observed in pancreatic cancer patients[15], elucidating interactions among *GP2* variants, *KRAS* oncogenic mutations, and other potential effector genes would provide insights into pancreatic carcinogenesis.

Previous epidemiological studies have suggested that HbA1c levels, even in nondiabetic ranges, or changes in HbA1c levels in new-onset T2D are associated with pancreatic cancer risk[25,26]. Our MR analysis of selected metabolic factors provided corroborating evidence that HbA1c genetic index levels may be associated with pancreatic cancer risk. This result was also partially consistent with a previous MR analysis, in which T2D was not implicated but the genetic indices of BMI and fasting insulin were associated with pancreatic cancer[27]. However, it should be noted that the MR results may be influenced by a few SNPs with relatively large effect sizes. To address this possibility, we further applied MR-PRESSO and detected two outlying SNPs (Supplementary Data 6), one of which was the *GP2* SNP. This finding indicated that the *GP2* SNP may differ from other T2D-related

SNPs in terms of the effect on pancreatic cancer risk. It is likely that the null findings for the MR association for T2D reflect the phenotypic and genetic heterogeneity of T2D, but T2D may also be both a cause and consequence of pancreatic cancer[28]. Additional studies are needed to further explore the associations of T2D and metabolic factors with pancreatic cancer using the best available genetic instruments in the Japanese population.

Three genome-wide significant genes (*GP2*, *WNT2B*, and *KRT8*) emerged in the gene-based GWAS. Among these genes, *WNT2B* (1p13.1) showed suggestive evidence of association in the previous PanScan and PanC4 GWAS[6]. KRT8 belongs to a group of intermediate-filament cytoskeletal proteins involved in maintaining epithelial structural integrity[29]. KRT8 is expressed in both ductal and acinar single-layer epithelia, and mutations in the *KRT8* gene have been linked to exocrine pancreatic disorders and liver disease[30,31].

In conclusion, our GWAS meta-analysis identified a risk locus at chromosome 16p12.3 within the *GP2* gene for pancreatic cancer in populations of East Asian ancestry. Functional analyses using cell lines provided supporting evidence of the effect of the lead SNP rs78193826 of the *GP2* gene on KRAS activity. Further fine mapping and functional characterization are needed to elucidate the associations of common *GP2* gene variants with pancreatic cancer susceptibility. Our findings also highlight genetic susceptibility factors shared between T2D and pancreatic cancer.

## Methods

**Study samples**. We performed a GWAS meta-analysis based on three Japanese studies: the Japan Pancreatic Cancer Research (JaPAN) consortium GWAS, the National Cancer Center (NCC) GWAS, and the BioBank Japan (BBJ) GWAS. An overview of the characteristics of the study populations is provided in Supplementary Table 1.

**JaPAN consortium GWAS**. Participants in this GWAS were drawn from the JaPAN consortium[32]. Two case–control datasets were combined, resulting in data from a total of 945 pancreatic cancer patients and 3134 controls. The vast majority of patients were diagnosed with primary adenocarcinoma of the exocrine pancreas (ICD-O-3 codes C250–C259). The first dataset included 622 pancreatic cancer patients who were recruited from January 2010 to July 2014 at 5 participating hospitals in the Central Japan, Kanto, and Hokkaido regions. This multi-institutional case–control study collected questionnaire data on demographic and lifestyle factors and 7-ml blood samples from the study participants. The second dataset included 323 patients with newly diagnosed pancreatic cancer and 3134 control subjects recruited for an epidemiological research program at Aichi Cancer Center (HERPACC) between 2005 and 2012. All outpatients on their first visit to Aichi Cancer Center were invited to participate in HERPACC. Those who agreed to participate completed a self-administered questionnaire and provided a 7-ml blood sample. After quality control, 943 cases and 3057 controls remained for the subsequent analysis (Supplementary Data 1). None of the control subjects had a diagnosis of cancer by the time of recruitment. Written informed consent was obtained from all study participants, and the study protocol was approved by the Ethical Review Board of Aichi Medical University, the Institutional Ethics Committee of Aichi Cancer Center, the Human Genome and Gene Analysis Research Ethics Committee of Nagoya University, and the ethics committees of all participating hospitals.

**BioBank Japan GWAS**. Pancreatic cancer patient data were obtained from the BBJ GWAS, which was launched in 2003 and collected DNA and clinical information from approximately 200,000 patients, including those with pancreatic cancer[33]. Overall, 422 pancreatic cancer patients with available genotype data were recruited from 2003 to 2008. Clinical information was collected using a standardized questionnaire. This study was approved by the ethics committees of the RIKEN Center for Integrative Medical Sciences. Controls were drawn from the participants in four population-based cohort studies in Japan: the Japan Multi-Institutional Collaborative Cohort Study, the JPHC, the Tohoku Medical Megabank Project Organization, and the Iwate Tohoku Medical Megabank Organization (Supplementary Note 2). In total, 28,870 controls who passed genotype data quality-control assessments were included in the study. In all participating cohort studies, informed consent was obtained from the participants by following the protocols approved by the corresponding institutional ethics committees. Detailed descriptions of the BBJ and each cohort study are provided in Supplementary Note 2.

**National Cancer Center GWAS**. The case and control samples were derived from a previous pancreatic cancer GWAS[11]. Case subjects were 677 patients diagnosed with invasive pancreatic ductal adenocarcinoma at the NCC Hospital, Tokyo, Japan. Controls consisted of 677 Japanese volunteers who participated in a health check-up program in Tokyo. After preimputation quality control, data from 674 cases and 674 controls remained for the subsequent analysis (Supplementary Data 1). This project was approved by the ethics committee of the NCC.

**Quality control and genotype imputation**. Quality control for samples and SNPs was performed based on the study-specific criteria. For the study that included data genotyped on two different platforms, we performed imputation using those SNPs that were available from both genotyping platforms. Genotype data in each study were imputed separately based on the 1000 Genomes Project reference panel (Phase 3, all ethnicities). The phasing was performed with the use of SHAPEIT (v2)[34] or Eagle2 (v2)[35], and the imputation was performed using minimac3[36] or IMPUTE (v2)[37]. Information on the study-specific genotyping, imputation, and analysis tools is provided in Supplementary Data 1. After genotype imputation, quality control was applied to each study. SNPs with an imputation quality of $r^2 < 0.5$ or an MAF of <0.01 were excluded. SNPs that passed quality control in at least two cohorts were included in the meta-analysis.

**Association analysis for SNPs and pancreatic cancer**. The association of pancreatic cancer with the SNP allele dose was estimated using logistic regression analysis with adjustment for the first two principal components. The association magnitudes and standard errors were used in the subsequent meta-analysis.

**GWAS meta-analysis**. We performed a meta-analysis of three pancreatic cancer GWASs (JaPAN, BBJ, and NCC). The association results for each SNP across the studies were combined with the METAL software (v2011-03-25) in a fixed effects IVW meta-analysis. Heterogeneity in allelic associations was assessed using the $I^2$ index. The meta-analysis included 7,914,378 SNPs with genotype data available from at least two cohorts. A P value threshold of $5 \times 10^{-8}$ was used to establish a threshold for genome-wide significance. We assessed the inflation of test statistics using the genomic control lambda.

**Replication analysis**. We sought replication of the SNP associations in six additional studies involving populations of East Asian origin, including Japanese, Japanese American, and Chinese individuals. In total, we assembled genotype data from 1926 cases and 8896 controls for the replication analysis. Detailed information on the study descriptions, quality-control thresholds, and exclusion criteria for each replication cohort is provided in Supplementary Data 1 and the Supplementary Note 3. The association between the SNP allele dose and pancreatic cancer risk in each replication cohort was estimated using logistic regression analysis with the adjustment for study-specific covariates shown in Supplementary Data 1. For the lead SNP rs78193862, we also performed a fixed-effects IVW meta-analysis of SNP associations by combining all six study sample sets included in the replication analysis.

**Functional annotation**. To prioritize the associated SNPs of the identified loci, we adopted a series of bioinformatic approaches to collate functional annotations. We first used ANNOVAR[38] to obtain an aggregate set of functional annotations—including gene locations and impacts of amino acid substitutions based on prediction tools, such as SIFT, PolyPhen-2, and CADD—for SNPs with P values <5 × $10^{-8}$ for pancreatic cancer. We also explored potential effects on gene regulation by annotating these SNPs with information from the RegulomeDB database[39].

**Functional characterization**. We chose PaTu 8988s for the functional study, because it is the only pancreatic adenocarcinoma cell line expressing *GP2* according to the Cancer Cell Line Encyclopedia. PaTu 8988s cells were obtained from the Deutsche Sammlung von Mikroorganismen und Zellkulturen (DSMZ, Braunschweig, Germany) and maintained with RPMI 1640 (Invitrogen) plus 10% fetal bovine serum at 37 °C in a 5% $CO_2$ cell culture incubator. PaTu 8988s cells were genotyped for identity at BEX CO., LTD. and tested routinely for mycoplasma contamination.

**Generation of the GP2_V282M PaTu 8988s cell line**. Mutation was induced through CRISPR-Cas9-mediated homologous recombination. The plasmid pSpCas9(BB)-2A-GFP (PX458) (Addgene plasmid # 48138) was purchased from Addgene (Cambridge, MA)[40]. To generate a nucleotide mutation from G to A at codon 282 (synonymous with 285, 429, and 432; Supplementary Data 2) of the *GP2* gene, we selected an single-guide RNA target with CHOPCHOP (https://chopchop.cbu.uib.no)[41] and designed a DNA repair template with 45 bp homology arms. Oligonucleotide pairs were annealed and ligated into the BbsI-linearized PX458 plasmid. Cells were transfected with the vector and the repair template using Lipofectamine 3000 (Life Technologies) according to the manufacturer's instructions. To obtain monoclonal clones, GFP-positive cells were sorted as single cells into 96-well plates using a BD FACSAria III cell sorter (BD Biosciences) 48 h post-transfection. After 3 weeks of culture, cells were distributed into two 24-well plates,

followed by Sanger sequencing-based genotyping. Mutation was also confirmed by RNA-seq. A clone harboring the precise mutation was used for further analysis. All oligonucleotide primers were obtained from FASMAC (Kanagawa, Japan; Supplementary Table 7). Control cell lines were obtained from clones without induced mutations at codon 282 by the same plasmid transfection.

**Transient transfection experiments.** In all, $2 \times 10^5$ Patu-S cells plated in 6-well plate were transfected with the empty vector alone or the sgGP2-cloned plasmid and the repair template using Lipofectamine 3000 (Life Technologies) according to the manufacturer's instructions. Total RNA was isolated 72 h post-transfection followed by cDNA synthesis and qRT-PCR.

**RNA isolation and cDNA synthesis.** Total RNA was isolated using the miRNeasy Mini Kit (Qiagen) with DNase I (Qiagen) digestion according to the manufacturer's instructions. cDNA was synthesized from total RNA using Superscript III (Invitrogen) and random primers (Invitrogen).

**Quantitative real-time PCR.** qRT-PCR was performed using SYBR® Premix Ex Taq TM (TaKaRa) on an Applied Biosystems 7900HT Real-Time PCR System. All oligonucleotide primers were obtained from FASMAC (Kanagawa, Japan; Supplementary Table 7).

**RNA-sequencing.** RNA-seq was performed by GeneWiz Inc. (Saitama, Japan) in paired-end mode. RNA-seq reads were mapped to NCBI37 with TopHat2 and quantified to the human transcriptome (refGene) with GFOLD[42]. Gene expression was quantified in the form of RPKM (reads per kilobase of transcript, per million mapped reads). The reference sequence and refGene GTF files were obtained from iGenomes.

**Differential expression analysis and GSEA.** Differentially expressed genes were identified using GFOLD (v1.1)[42] based on comparisons between the GP2_V282M and GP2_WT groups. Significantly differentially expressed genes were defined as genes with empirical false discovery rates (FDR) < 0.10. The GSEA was performed using the MSigDB v6.2 collections—H and C6[14]. Genes with coding lengths <200 bp or <10 mean reads in either the GP2_V282M or GP2_WT groups were excluded. A rank list was generated by ordering each gene according to the log2-fold-change value (log2fdc from GFOLD), which was calculated from the expression level ratio of GP2_V282M/GP2_WT. These rank lists were used in a weighted, preranked GSEA. Sets of 1000 permutations of the genes were applied in the preranked GSEA performed with the above-described collections of gene sets. An FDR < 0.10 was considered significant for the GSEA analysis.

**MR analysis.** We performed MR analyses for the associations of pancreatic cancer with selected metabolic traits, including T2D, HbA1c, blood glucose, and BMI. As genetic instruments for each trait, we selected genome-wide significant SNPs that had been reported in the three previously published GWAS meta-analyses involving Japanese subjects[17,18,43]. For the two-sample MR analysis of T2D and pancreatic cancer, we did not exclude duplicate samples (15.5% found only in the controls) because retaining these samples was unlikely to introduce substantial bias[44]. Data from 106 pancreatic cancer cases were excluded from the HbA1c GWAS, and the association magnitudes for the HbA1c-associated SNPs were re-estimated. After the exclusion of SNPs on the X chromosome or SNPs without genotype data (Supplementary Data 6–9), the summary data were available for 82 T2D-related SNPs, 25 HbA1c-related SNPs, 15 blood glucose-related SNPs, and 76 BMI-related SNPs. The associations of these SNPs with pancreatic cancer risk were analyzed using the IVW and MR–Egger regression methods. MR analysis was performed with the MendelianRandomization package[45]. Given that the presence of horizontal pleiotropy may violate MR assumptions, leading to invalid results, we further applied the MR-PRESSO test to detect and correct for horizontal pleiotropic outliers[46].

**Gene-based analysis.** SNP-based $P$ values for 7,914,378 SNPs were combined into gene-based $P$ values for 17,581 genes using the MAGMA software version 1.06[18]. SNP summary statistics ($P$ values) from the meta-analysis were used as input for MAGMA. In gene-based association statistics, LD between SNPs was accounted for, and the $P$ value threshold for genome-wide significant associations was set at $2.84 \times 10^{-6}$ (=0.05/17,581). The 1000 Genomes reference panel (Phase 3, East Asian) was used to control for LD. We did not include any upstream/downstream regions around the genes in this analysis; only variants located between the first exon and the last exon of a gene were used to calculate the gene-based $P$ values. The NCBI Gene database was used to define genomic intervals for protein-coding genes. To replicate the association between *KRT8* and pancreatic cancer, we applied SNP summary statistics from PanScan 1 and PanScan 2 (pha002889.1)[13] to MAGMA. The MAF of the Haplotype Map (HapMap) project Phase 2 CEU samples for each SNP was added to the summary statistics, because the pha002889.1 data did not include the MAFs. We excluded variants with call fractions <95% in either the case or control data, Hardy–Weinberg equilibrium $P$ value $<10^{-6}$ in the controls, or MAF < 0.01. The 1000 Genomes reference panel

(Phase 3, European) was used to control for LD. The significance level was set at $\alpha = 0.05$.

**Reporting summary.** Further information on research design is available in the Nature Research Reporting Summary linked to this article.

## Data availability

The summary GWAS statistics for this analysis are available on the website of the JaPAN consortium [http://www.aichi-med-u.ac.jp/JaPAN/current_initiatives-e.html]. The RNA-seq data are available at the Gene Expression Omnibus under the following accession number: GSE147368. The other datasets generated during this study are available from the corresponding author upon reasonable request. The source data underlying Fig. 4e are provided as a Source data file.

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

## Acknowledgements

We thank Mayuko Masuda, Kikuko Kaji, Kazue Ando, Etsuko Ohara, Sumiyo Asakura, Keiko Hanai, and Yoko Mitsuda for assistance with data collection. We would like to express our gratefulness to the staff of BioBank Japan for their outstanding assistance. This work was supported by the Ministry of Health, Labor, and Welfare of Japan (H21-11-1) and the Ministry of Education, Culture, Sports, Science, and Technology of Japan (Nos. 16H06277, 17K09095, 17H04127, 26293145, 26253041, and 17015018). HER-PACC, a part of the JaPAN consortium, was supported by Grants-in-Aid for Scientific Research for Priority Areas of Cancer (No. 17015018) and Innovative Areas (No. 221S0001); JSPS KAKENHI grants (Nos. 26253041, 15H02524, 16H06277, and 18H03045) from the Japanese Ministry of Education, Science, Sports, Culture and Technology; and a Grant-in-Aid for the Third Term Comprehensive 10-year Strategy for Cancer Control from the Ministry of Health, Labor and Welfare of Japan and the Cancer Biobank Aichi. This study was partially supported by the BioBank Japan project and the Tohoku Medical Megabank project, which is supported by the Ministry of Education, Culture, Sports, Sciences and Technology of Japan and the Japan Agency for Medical Research and Development. The JPHC Study has been supported by the National Cancer Center Research and Development Fund (23-A31[toku], 26-A-2, and 29-A-4) since 2011 and was supported by a Grant-in-Aid for Cancer Research from the Ministry of Health, Labor and Welfare of Japan from 1989 to 2010. The case–cohort study within the JPHC Study was supported by the National Cancer Center Research and Development Fund (28-A-19 and 31-A-18) and by Practical Research for Innovative Cancer Control (JP16ck0106095h0003 and JP19ck0106266h0003) from the Japan Agency for Medical Research and Development, AMED. The J-MICC study was supported by Grants-in-Aid for Scientific Research for Priority Areas of Cancer (No. 17015018) and Innovative Areas (No. 221S0001) and a JSPS KAKENHI grant (No. 16H06277) from the Japanese Ministry of Education, Science, Sports, Culture and Technology. The JEPA study was supported by Grants-in-Aid for Scientific Research (S) and a JSPS KAKENHI grant (No. 15H05791) from the Japanese Ministry of Education, Science, Sports, Culture and Technology. The Yale-Shanghai study was supported by NIH/NCI 5R01 CA114421. This work utilized the computational resources of the NIH HPC Biowulf cluster (http://hpc.nih.gov). The MEC replication study was supported by NIH/NCI CA209798, and CA164973.

## Author contributions

Y.L., M.N., Y. Hosono, H. Ito, A.I., M.O., T.O., S. Kamiya, and C.W. designed the study. Y.L., M.N., Y. Hosono, and Y. Kamatani wrote the manuscript. M.N., F.K., Y. Kobayashi., Y. Kamatani., M.A., K.I., and J.Z. performed the statistical analysis. H.S., H.I., M.O., T.S., M. Matsuyama., N.S., M.M., S. Kobayashi, T.F., M.U., S.O., N.E., S. Kuruma, Mitsura Mori., H.N., Y.A., K.H., Y.S., Y.M., K. Matsuda, M. Hirata, K. Shimada, T.O., Y.W., K. Kuriki, A.K., R.O., H.M., T.T., S.S., K.W., T. Yamaji, M.I., N.S., A.G., S.T., K. Kinoshita, N.F., F.K., A.S., S.S.N., K.T., K. Suzuki, Y.O., M. Horikoshi, T. Yamauchi, T. Kadowaki, H.Y., L.T.A., Y.D., H. Ishii, H.E., D.B., C.A.H., L.L.M., Masaki Mori, H.R., V.W.S., T. Yoshida, and M.K. contributed to data collection. Y. Hosono, A.I., T.M., Y. Hayashi, H.E., T. Kohmoto, I.I., Y. Kasugai, M.K., T. Kawaguchi, M.T., and F.M. contributed to SNP genotyping and functional characterization. S. Kikuchi and K. Matsuo supervised the study. All authors approved the final version of the manuscript.

## Competing interests

The authors declare no competing interests.

## Additional information

Yingsong Lin [1,60✉], Masahiro Nakatochi [2,3,60✉], Yasuyuki Hosono [4,60], Hidemi Ito [5,6,60], Yoichiro Kamatani [7,8], Akihito Inoko [9,10], Hiromi Sakamoto[11], Fumie Kinoshita[12], Yumiko Kobayashi[12], Hiroshi Ishii[13], Masato Ozaka[14], Takashi Sasaki[14], Masato Matsuyama[14], Naoki Sasahira[14], Manabu Morimoto[15], Satoshi Kobayashi[15], Taito Fukushima[15], Makoto Ueno [15], Shinichi Ohkawa[15], Naoto Egawa[16], Sawako Kuruma[17], Mitsuru Mori[18], Haruhisa Nakao[19], Yasushi Adachi [20], Masumi Okuda[21], Takako Osaki [22], Shigeru Kamiya[22], Chaochen Wang [1], Kazuo Hara[23], Yasuhiro Shimizu[24], Tatsuo Miyamoto [25], Yuko Hayashi[4], Hiromichi Ebi[4], Tomohiro Kohmoto[26,27], Issei Imoto[27], Yumiko Kasugai [9,28], Yoshinori Murakami [29], Masato Akiyama[7,30], Kazuyoshi Ishigaki [7], Koichi Matsuda [31], Makoto Hirata [29], Kazuaki Shimada[32], Takuji Okusaka [33], Takahisa Kawaguchi[34], Meiko Takahashi[34], Yoshiyuki Watanabe[35], Kiyonori Kuriki[36], Aya Kadota[37], Rieko Okada [38], Haruo Mikami[39], Toshiro Takezaki[40], Sadao Suzuki[41], Taiki Yamaji[42], Motoki Iwasaki[42], Norie Sawada [42], Atsushi Goto[42], Kengo Kinoshita [43], Nobuo Fuse [43], Fumiki Katsuoka[43], Atsushi Shimizu [44], Satoshi S. Nishizuka[44], Kozo Tanno[44,45], Ken Suzuki [7,46,47,48], Yukinori Okada [7,48,49], Momoko Horikoshi[47], Toshimasa Yamauchi [46], Takashi Kadowaki [46], Herbert Yu [50], Jun Zhong [51], Laufey T. Amundadottir [51], Yuichiro Doki[52], Hideshi Ishii[53], Hidetoshi Eguchi[52], David Bogumil [54], Christopher A. Haiman[54,55], Loic Le Marchand[50], Masaki Mori[56], Harvey Risch[57], Veronica W. Setiawan[54,55], Shoichiro Tsugane [58], Kenji Wakai[38], Teruhiko Yoshida[11], Fumihiko Matsuda[34], Michiaki Kubo [59], Shogo Kikuchi[1] & Keitaro Matsuo [9,28✉]

[1]Department of Public Health, Aichi Medical University School of Medicine, Nagakute, Aichi 480-1195, Japan. [2]Division of Public Health Informatics, Department of Integrated Health Sciences, Nagoya University Graduate School of Medicine, Nagoya 461-8673, Japan. [3]Department of Nursing, Nagoya University Graduate School of Medicine, Nagoya 461-8673, Japan. [4]Division of Molecular Therapeutics, Aichi Cancer Center Research Institute, Nagoya 464-8681, Japan. [5]Division of Cancer Information and Control, Aichi Cancer Center Research Institute, Nagoya 464-8681, Japan. [6]Department of Descriptive Epidemiology, Nagoya University Graduate School of Medicine, Nagoya 466-8550, Japan. [7]Laboratory for Statistical Analysis, RIKEN Center for Integrative Medical Sciences, Yokohama 230-0045, Japan. [8]Laboratory of Complex Trait Genomics, Department of Computational Biology and Medical Sciences, Graduate School of Frontier Sciences, The University of Tokyo, Tokyo 108-8639, Japan. [9]Division of Cancer Epidemiology and Prevention, Aichi Cancer Center Research Institute, Nagoya 464-8681, Japan. [10]Department of Pathology, Aichi Medical University School of Medicine, Nagakute 480-1195, Japan. [11]Genetics Division, National Cancer Center Research Institute, Tokyo 104-0045, Japan. [12]Data Science Division, Data Coordinating Center, Department of Advanced Medicine, Nagoya University Hospital, Nagoya 461-8673, Japan. [13]Chiba Cancer Center, Chiba 260-8717, Japan. [14]Department of Hepato-biliary-pancreatic Medicine, The Cancer Institute Hospital of Japanese Foundation for Cancer Research, Tokyo 135-8550, Japan. [15]Department of Gastroenterology, Hepatobiliary and Pancreatic Medical Oncology Division, Kanagawa Cancer Center, Yokohama 241-8515, Japan. [16]Department of Gastroenterology, Tokyo Metropolitan Hiroo Hospital, Tokyo 150-0013, Japan. [17]Department of Gastroenterology, Tokyo Metropolitan Komagome Hospital, Tokyo 113-8677, Japan. [18]Hokkaido Chitose College of Rehabilitation, Hokkaido 066-0055, Japan. [19]Division of Hepatology and Pancreatology, Aichi Medical University School of Medicine, Nagakute 480-1195, Japan. [20]Sapporo Shirakabadai Hospital, Sapporo 062-0052, Japan. [21]Department of Pediatrics, Hyogo College of Medicine, Nishinomiya, Hyogo 663-8501, Japan. [22]Department of Infectious Diseases, Kyorin University School of Medicine, Tokyo 181-8611, Japan. [23]Department of Gastroenterology, Aichi Cancer Center Hospital, Nagoya 464-8681, Japan. [24]Department of Gastroenterological Surgery, Aichi Cancer Center Hospital, Nagoya 464-8681, Japan. [25]Department of Genetics and Cell Biology, Research Institute for Radiation Biology and Medicine, Hiroshima University, Hiroshima 734-8553, Japan. [26]Department of Human Genetics, Tokushima University Graduate School of Medicine, Tokushima 770-8503, Japan. [27]Division of Molecular Genetics, Aichi Cancer Center Research Institute, Nagoya 464-8681, Japan. [28]Department of Cancer Epidemiology, Nagoya University Graduate School of Medicine, Nagoya 466-8550, Japan. [29]Division of Molecular Pathology, Institute of Medical Science, The University of Tokyo, Tokyo 108-8639, Japan. [30]Department of Ophthalmology, Graduate School of Medical Sciences, Kyushu University, Fukuoka 812-8582, Japan. [31]Graduate School of Frontier Sciences, The University of Tokyo, Tokyo 108-8639, Japan. [32]Department of Hepatobiliary and Pancreatic Surgery, National Cancer Center Hospital, Tokyo 104-0045, Japan. [33]Department of Hepatobiliary and Pancreatic Oncology, National Cancer Center Hospital, Tokyo 104-0045, Japan. [34]Center for Genomic Medicine, Graduate School of Medicine, Kyoto University, Kyoto 606-8501, Japan. [35]Department of Epidemiology for Community Health and Medicine, Kyoto Prefectural University of Medicine, Kyoto 602-8566, Japan. [36]Laboratory of Public Health, School of Food and Nutritional Sciences, University of Shizuoka, Shizuoka 422-8526, Japan. [37]Department of Public Health, Shiga University of Medical Science, Otsu 520-2192, Japan. [38]Department of Preventive Medicine, Nagoya University Graduate School of Medicine, Nagoya 466-8550, Japan. [39]Cancer Prevention Center, Chiba Cancer Center Research Institute, Chiba 260-8717, Japan. [40]Department of International Island and Community Medicine, Kagoshima University Graduate School of Medical and Dental Sciences, Kagoshima 890-8544, Japan. [41]Department of Public Health, Nagoya City University Graduate School of Medical Sciences, Nagoya 467-8601, Japan. [42]Division of Epidemiology, Center for Public Health Sciences, National Cancer Center, Tokyo 104-0045, Japan. [43]Tohoku Medical Megabank Organization, Tohoku University, Sendai 980-8573, Japan. [44]Iwate Tohoku Medical Megabank Organization, Iwate Medical University, Iwate 028-3694, Japan. [45]Department of Hygiene and Preventive Medicine, School of Medicine, Iwate Medicalm University, Iwate 028-3694, Japan. [46]Department of Diabetes and Metabolic Diseases, Graduate School of Medicine, The University of Tokyo, Tokyo 113-0033, Japan. [47]Laboratory for Endocrinology, Metabolism and Kidney Diseases, RIKEN Centre for Integrative Medical Sciences, Yokohama 230-0045, Japan. [48]Department of Statistical Genetics, Osaka University Graduate School of Medicine, Osaka 565-0871, Japan. [49]Laboratory of Statistical Immunology, Immunology Frontier Research Center (WPI-IFReC), Osaka University, Osaka 565-0871, Japan. [50]University of Hawaii Cancer Center, Honolulu, HI 96813, USA. [51]Laboratory of Translational Genomics, Division of Cancer Epidemiology and Genetics, National Cancer Institute, National Institutes of Health, Bethesda, MD 20892, USA. [52]Department of Gastroenterological Surgery,

Graduate School of Medicine, Osaka University, Osaka 565-0871, Japan. [53]Department of Medical Data Science, Graduate School of Medicine, Osaka University, Osaka 565-0871, Japan. [54]Department of Preventive Medicine, Keck School of Medicine, University of Southern California, Los Angeless, CA 90033, USA. [55]Norris Comprehensive Cancer Center, University of Southern California, Los Angeles, CA 90033, USA. [56]Department of Surgery and Science, Graduate School of Medical Sciences, Kyushu University, Fukuoka 812-8582, Japan. [57]Department of Chronic Disease Epidemiology, Yale School of Public Health, New Haven, CT 06520, USA. [58]Center for Public Health Sciences, National Cancer Center, Tokyo 104-0045, Japan. [59]RIKEN Center for Integrative Medical Sciences, Yokohama 230-0045, Japan. [60]These authors contributed equally: Yingsong Lin, Masahiro Nakatochi, Yasuyuki Hosono, Hidemi Ito. ✉email: linys@aichi-med-u.ac.jp; mnakatochi@met.nagoya-u.ac.jp; kmatsuo@aichi-cc.jp

