## [Peer Review File · Nature Communications]

Reviewers' comments:

Reviewer #1 (Remarks to the Author):

This analysis reports on a mixture of previously published and novel genome-wide data for pancreatic cancer in Japan to report on a novel association with genetic variants in 16p12.3 that locate to the glycoprotein 2 (GP2) gene. Further replication of these variants in an independent series replicate the finding, if at a lower odds ratio. There are interesting implications for the role of this gene in the etiology of pancreatic cancer.

The genetic link between pancreatic cancer and other metabolic factors has been illustrated in the European data based on Panscan, with mendelian randomization analysis illustrating the probable causal effect of obesity, insulin resistance, and possible beta cell function. It would therefore be interesting to expand the mendelian randomization analysis to consider these aspects in the Japanese population, assuming that the genetic instruments are available.

The authors should also consider conducting a two stage Mendelian randomization analysis, allowing for risk estimates of the effect of changes in HbA1c on pancreatic cancer risk, as well as other metabolic factors including obesity and insulin resistance.

Reviewer #2 (Remarks to the Author):

1) greater detail should be provided on the imputation including reference panel, was imputation conducted separately for each array and then combined (joint imputation of all datasets can cause false signals

2) The risk allele frequency in cases varies considerably across populations in the replication cohorts, while the frequency in controls is consistent, this should be commented on.

3) important details are missing from table 1, including the RAF, and imputation quality. Also the imputation quality for each separate dataset and RAF should be provided.

4) Details on study population (case and control descriptive information age, gender etc) should be provides.

5) No controls including in expression results, these should be added.

6) MR results strongly influenced by a few SNPS, this should be discussed.

Comment: The Childs et al paper cited was both the PanC4 consortium and PanScan please correct this in the text.

Reviewers' comments:

Reviewer #1 (Remarks to the Author):

This analysis reports on a mixture of previously published and novel genome-wide data for pancreatic cancer in Japan to report on a novel association with genetic variants in 16p12.3 that locate to the glycoprotein 2 (GP2) gene. Further replication of these variants in an independent series replicate the finding, if at a lower odds ratio. There are interesting implications for the role of this gene in the etiology of pancreatic cancer.

The genetic link between pancreatic cancer and other metabolic factors has been illustrated in the European data based on Panscan, with mendelian randomization analysis illustrating the probable causal effect of obesity, insulin resistance, and possible beta cell function. It would therefore be interesting to expand the mendelian randomization analysis to consider these aspects in the Japanese population, assuming that the genetic instruments are available.

The authors should also consider conducting a two stage Mendelian randomization analysis, allowing for risk estimates of the effect of changes in HbA1c on pancreatic cancer risk, as well as other metabolic factors including obesity and insulin resistance.

Reviewer #2 (Remarks to the Author):

1) greater detail should be provided on the imputation including reference panel, was imputation conducted separately for each array and then combined (joint imputation of all datasets can cause false signals

2) The risk allele frequency in cases varies considerably across populations in the replication cohorts, while the frequency in controls is consistent, this should be commented on.

3) important details are missing from table 1, including the RAF, and imputation quality. Also the imputation quality for each separate dataset and RAF should be provided.

4) Details on study population (case and control descriptive information age, gender etc) should be provides.

5) No controls including in expression results, these should be added.

6) MR results strongly influenced by a few SNPS, this should be discussed.

Comment: The Childs et al paper cited was both the PanC4 consortium and PanScan please correct this in the text.

Response to Reviewer #1's comments

This analysis reports on a mixture of previously published and novel genome-wide data for pancreatic cancer in Japan to report on a novel association with genetic variants in 16p12.3 that locate to the glycoprotein 2 (GP2) gene. Further replication of these variants in an independent series replicate the finding, if at a lower odds ratio. There are interesting implications for the role of this gene in the etiology of pancreatic cancer.

We thank the reviewer for the positive comments regarding our study.

The genetic link between pancreatic cancer and other metabolic factors has been illustrated in the European data based on Panscan, with mendelian randomization analysis illustrating the probable causal effect of obesity, insulin resistance, and possible beta cell function. It would therefore be interesting to expand the mendelian randomization analysis to consider these aspects in the Japanese population, assuming that the genetic instruments are available.

We agree that it would be interesting to expand the Mendelian (MR) analyses in the Japanese population. Accordingly, we conducted additional MR analyses for body mass index (BMI) and blood glucose using associated genetic variants (SNPs) that were identified in the GWASs involving Japanese subjects. No significant associations were noted for either trait. For insulin resistance, which was reported in the PanScan study, we did not perform MR analyses because of relatively insufficient data on associated genetic variants in the Japanese population. We have added the results of BMI and blood glucose to the main text and Supplementary Figure 5.

We also conducted MR analyses of other metabolic factors, including cholesterol, high-density lipoprotein (HDL), low-density lipoprotein (LDL), and triglycerides, using

the genome-wide significant SNPs for each trait that were available from the GWAS by Kanai et al. (Nature Genetics 2017). No significant associations between genetically determined levels for each trait and pancreatic cancer risk were observed (data not shown). Additional studies, however, are needed to further explore the associations of T2D and metabolic factors with pancreatic cancer using the best-available genetic instruments in the Japanese population.

Page 11

Among 15 blood glucose-related SNPs, rs4581570 at 13q12.2 (PDX1) was significant after Bonferroni correction ($P < 0.0033$, Supplementary Data 7). In addition, none of the 25 HbA1c-associated (Supplementary Data 8) and 76 body mass index (BMI)-associated SNPs (Supplementary Data 9) were significant in our GWAS meta-analysis.

Page 12

In addition, we found no significant associations between genetic index levels for the other two metabolic factors (blood glucose and BMI) and pancreatic cancer risk (Supplementary Figure 5).

Page 16 Discussion

Our MR analysis of selected metabolic factors provided corroborating evidence that HbA1c genetic index levels may be associated with pancreatic cancer risk.

Page 16 Discussion

Additional studies are needed to further explore the associations of T2D and metabolic factors with pancreatic cancer using the best-available genetic instruments in the Japanese population.

The authors should also consider conducting a two stage Mendelian randomization analysis, allowing for risk estimates of the effect of changes in HbA1c on pancreatic cancer risk, as well as other metabolic factors including obesity and insulin resistance.

To the best of our knowledge, the two-stage Mendelian randomization (MR) approach, like the two-stage least squares method, is based on individual-level data in a one-sample setting, with genetic variants, risk factors and outcomes all measured in the same participants. (Burgess, S., et al. A review of instrumental variable estimators for

Mendelian randomization. *Stat Methods Med Res* 26(5): 2333-2355 (2017)). According to Burgess et al., although estimates from IV analysis in a one-sample setting are asymptotically unbiased, they can have substantial bias in finite samples. The causal estimate from a one-sample analysis with weak instrumental variables may be biased in the direction of the observational association between the risk factor and outcome (Burgess, S., Bias due to participant overlap in two-sample Mendelian randomization. *Genetic Epidemiol* 2016). With the increasing availability of summarized instrumental variables, a two-sample analysis strategy is more frequently adopted in MR analyses because it is less biased, and any bias is in the direction of the null.

We appreciate the reviewer's suggestion regarding conducting a two-stage MR analysis. Unfortunately, we were not able to perform a two-stage MR because of the lack of individual-level data on the history of T2D or T2D-related traits in a one-sample setting. In light of the limitations and strengths of two-stage and two-sample MR, as well as data availability, we conducted additional two-sample MR analyses and presented the estimated effect sizes in Figure 5 and Supplementary Figures 4 and 5.

Response to Reviewer #2's comments

1) greater detail should be provided on the imputation including reference panel, was imputation conducted separately for each array and then combined (joint imputation of all datasets can cause false signals).

We agree that presenting details on the imputation method and quality is important for judging the validity of our study results. One potential limitation is that joint imputation was performed for SNPs genotyped with seemingly different arrays (Illumina HumanHap550 and Illumina Human610-Quad) in the NCC GWAS; however, these two platforms can be considered the same array because the latter contained 100% of the contents of the former. Moreover, only those genotyped SNPs available from both arrays were used in the imputation. To further address this issue, we conducted a principal component analysis (PCA), demonstrating that the PCA plot for the NCC SNP genotype data did not differ between the two arrays and that genetic inflation was not observed ($\lambda_{GC}=1.01$) (Figure). In addition to the NCC GWAS, essentially the same array was used in the individual GWAS. For these reasons, we consider that the imputation methods adopted in our study were unlikely to have introduced serious bias. We have added relevant descriptions to the manuscript as follows.

Figure. PCA plot for the NCC SNP genotype data

Page 19 Methods

Quality control for samples and SNPs was performed based on the study-specific criteria. For the study that included data genotyped on two different platforms, we performed imputation using those SNPs that were available from

both genotyping platforms. Genotype data in each study were imputed separately based on the 1000 Genomes Project reference panel (Phase 3, all ethnicities). The phasing was performed with the use of SHAPEIT234 or Eagle235, and the imputation was performed using minimac336 or IMPUTE237. Information on the study-specific genotyping, imputation, and analysis tools is provided in Supplementary Data 1. After genotype imputation, quality control was applied to each study. SNPs with an imputation quality of $r^2 < 0.5$ or an MAF of < 0.01 were excluded. SNPs that passed quality control in at least two cohorts were included in the meta-analysis.

2) The risk allele frequency in cases varies considerably across populations in the replication cohorts, while the frequency in controls is consistent, this should be commented on.

The reviewer raises an interesting point regarding a wide variation in risk allele

frequency in cases in the replication cohorts compared with control subjects. Assuming the accuracy of the imputation method and high-quality SNP data, we think that the main reason for the observed variations in cases may be due to random variation caused by a small sample size. As shown in **Supplementary Table 3**, the risk allele frequency was higher in controls than in cases in the JPHC study, generating an opposite direction in effect size from other replication cohorts. For the MEC study, cases and controls were Japanese Americans living in Hawaii and California in the United States. Population stratification is a possible reason, but the MEC results were adjusted for principal components. A further examination of the data suggested that other than in the JPHC and MEC replication cohorts, the risk allele frequency in cases did not vary considerably in other cohorts. In accordance with the reviewer's comment, we have added the following text to the revised manuscript.

Page 13

The MAF of rs78193826 in cases varies considerably across populations in the replication cohorts when compared with that in control subjects. The main reason may be due to random variation caused by a small sample size. As shown in Supplementary Table 3, the MAF was higher in controls than in cases in the JPHC study, generating an opposite direction in effect size from that observed in other replication cohorts. However, the overall positive association for rs78193826 was replicated in the analysis combining all replication cohorts. Another possible reason is population stratification, but the MEC results were adjusted for the principal component. After excluding the JPHC and MEC studies from the replication cohorts, we found no large variations in MAF in the cases.

3) important details are missing from table 1, including the RAF, and imputation quality. Also the imputation quality for each separate dataset and RAF should be provided.

We thank the reviewer for clarifying this point. Accordingly, we have added the RAFs for the cases and controls and the imputation quality scores for each dataset to the manuscript. Because Table 1 contains too many items to display, we rearranged it by moving the loci with suggestive significance to Supplementary Table 2. We also added imputation scores and RAFs to Supplementary Data 2 and Supplementary Table 3. Regarding the associations of selected SNPs with pancreatic cancer in the replication cohorts, we deleted Table 2 from the original manuscript and added the relevant information from Table 2 to Supplementary Table 3 because of the overlap in part of the results.

See Table 1, Supplementary Table 2, Supplementary Data 2, and Supplementary Table

3

4) Details on study population (case and control descriptive information age, gender etc) should be provides.

We thank the reviewer for this comment. Our understanding is that information on the study population, such as age and sex, has been provided in **Supplementary Table 1** in the original manuscript.

Supplementary Table 1

5) No controls including in expression results, these should be added.

We apologize for the confusion with regard to the definition of controls. Indeed, GP2_WT was used as a control cell line, which was obtained from clones in which the mutation at codon 282 was not induced by the same plasmid transfection. We have clarified this point by adding the description as follows.

Page 10

With RNA-sequencing (RNA-seq) analysis of two separate clones established from wild-type (GP2_WT, serving as a control group) and genome-edited (GP2_V282M) Patient 8988s cells, we demonstrated consistently different gene expression patterns between GP2_WT and GP2_V282M cells (Figure 4B, Supplementary Data 3).

Page 21

Control cell lines were obtained from clones without induced mutations at codon 282 by the same plasmid transfection.

0) MR results strongly influenced by a few SNPS, this should be discussed.

We agree with the reviewer that MR results are strongly influenced by a few SNPs with relatively large effects. For T2D, because we found evidence of possible horizontal pleiotropy in the MR-Egger analysis, we further applied MR-PRESSO and detected two outlying SNPs (Supplementary Data 6), one of which was the GP2 SNP. This finding indicated that the GP2 SNP may differ from other T2D-related SNPs in terms of the

effect on pancreatic cancer risk. We have added the following text to the Results and Discussion sections of the manuscript.

Page 11-12 Results

Inconsistent results were observed for I2D, with no significant association between the SNP-index I2D and pancreatic cancer risk based on the IVW method (Figure 5a). Although the MR-Egger analysis yielded a significant association, the intercept differed significantly from zero ($P < 0.05$, Supplementary Figure 4a), suggesting the presence of horizontal pleiotropy. Applying the MR Pleiotropy Residual Sum and Outlier (MR-PRESSO) test enabled the detection of two outlying SNPs (rs117267808 at 16p12.3 (GP2) and rs2290203 at 15q26.1 (PRC1-AS1)) (Supplementary Data 6). After correction for horizontal pleiotropy via outlier removal, the association remained nonsignificant in the IVW analysis ($\beta \pm SE = 0.08 \pm 0.06$, $P = 0.16$).

Page 16 Discussion

However, it should be noted that the MR results may be influenced by a few SNPs with relatively large effect sizes. To address this possibility, we further applied MR-PRESSO and detected two outlying SNPs (Supplementary Data 6), one of which was the GP2 SNP. This finding indicated that the GP2 SNP may differ from other I2D-related SNPs in terms of the effect on pancreatic cancer risk. It is likely that the null findings for the MR association for I2D reflect the phenotypic and genetic heterogeneity of I2D, but I2D may also be both a cause and consequence of pancreatic cancer.²⁸ Additional studies are needed to further explore the associations of I2D and metabolic factors with pancreatic cancer using the best-available genetic instruments in the Japanese population.

Page 22 Methods

Given that the presence of horizontal pleiotropy may violate MR assumptions, leading to invalid results, we further applied the MR-PRESSO test to detect and correct for horizontal pleiotropic outliers.⁴¹

7) Comment: The Childs et al paper cited was both the PanC4 consortium and PanScan please correct this in the text.

We thank the reviewer for this clarification. This has been corrected throughout the manuscript.

Page 7, Page 9, Page 12, Page 13, and Page 16

PanScan and PanC4 consortia

REVIEWERS' COMMENTS:

Reviewer #2 (Remarks to the Author):

The previous concerns have been addressed.

In the second round of peer review, Reviewer #2 mentioned in their confidential comments to the editor that vector control and potential off target p53 effects were not assessed in their functional analysis (figure 4). The authors have responded to this comment.

Reviewer #2's comment:

Regarding our response in the first revision that control cell lines are obtained from clones without induced mutation by the same plasmid transfection, Reviewer #2's is asking if we have looked at expression changes of WT GPT2 cells with the empty vector alone and at possible off target (i.e. p53) effects.

Response:

We appreciate the concern about the results of functional validation experiments. We did not look at changes of gene expression in clones established from WT GP2 cells transfected with the empty vector alone. Because the double strand breaks (DSBs) for genome editing occur through the transfection of the plasmid that we have used for genome editing in this study, we believe that off-target effects may not be induced by transfection of the empty vector alone, which cannot induce DSBs.

However, to address the reviewer #2's comments, we performed additional transient transfection experiments. First, we examined the mRNA expression changes in two p53 pathway genes (*CDKN1A* and *DDB2*) as well as TP53 itself under three different conditions: Patu-S parental cells, Patu-S cells transfected with empty vector alone, and Patu-S cells transfected with the plasmid that we have used for genome editing in this study. As shown in Supplementary Figure 9, no appreciable changes in the expression levels of those genes were noted among them. These findings suggest that possible off-target effects induced by the transfection of the empty vector as well as by DSBs do not affect the p53 pathway in our system, although we have used crude transfected cells in each group.

Additionally, we examined the mRNA expression changes in KRAS signaling pathway (*KLK7*, *BMPR1B*, and *KLK8*), under two different conditions: Patu-S parental cells and Patu-S cells transfected with the empty vector alone. No material changes in the expression levels of those genes were observed (Supplementary Figure 10). This additional experiment confirmed that possible off-target effects induced by the transfection of the empty vector alone do not affect KRAS signaling pathway, although we have used crude transfected cells in each group.

The reviewer #2 seems to be concerned about possible off-target effects induced by the CRISPR-Cas9 genome editing. We have used the clones without mutation induced by

transfection of the same genome editing plasmid as negative controls (GP2_WT). Using these clones as negative controls, we believe that we were able to see the GP2 mutation-specific gene expression changes in the clones of GP2_V425M, since CRISPR/Cas9-induced possible off-target effects (such as by unexpectedly induced

mutations or by the p53 pathway activation) may equally occur in the negative control clones and the GP2_V425M clones. To exclude the effects of clonal variation caused by selection bias, in addition, we have confirmed the expression changes of specific genes using 3 independent clones in each group in the original Figure 4E. Taken together, the data presented in the original paper (Figure 4E) exclude three possibilities: off-target effects through unexpectedly induced nonspecific mutations, non-specific p53 pathway activation through DSBs alone, and transcriptional changes of nonspecific genes through GP2 mutation-independent clonal variations.

Overall, we consider that the conclusion concerning the functional relevance of the lead SNP is reasonable, based on the results of additional experiments and those presented in the original paper. The revisions in the manuscript as follows.

Accordingly, we have added the following in the revised manuscript, with detailed results and discussions shown in Supplementary information.

Results Page 11

We also excluded the possibility of vector transfection–induced off-target effects or non-specific effects on the expression of genes in specific pathways (Supplementary Figures 9 and 10, Supplementary Discussion).

Methods Page 21

Transient transfection experiments were conducted to address the off-target effects that could be induced by the transfection of the vector (Supplementary Methods)

Details shown in Supplementary Information are as follows.

Supplementary Figure 9

Supplementary Figure 10

Supplementary Discussion

We did not look at changes of gene expression in clones established from WT GP2 cells transfected with the empty vector alone. Because the double strand breaks (DSBs) for genome editing occur through the transfection of the plasmid that we have used for genome editing in this study, we believe that off-target effects may not be induced by transfection of the empty vector alone, which cannot induce DSBs.

However, we performed additional transient transfection experiments to address the possible changes in gene expression of Patu-S cells transfected with the empty vector alone and off target effects. First, we examined the mRNA expression changes in two p53 pathway genes (CDKN1A and DDB2) as well as TP53 itself under three different conditions: Patu-S parental cells, Patu-S cells transfected with empty vector alone, and Patu-S cells transfected with the plasmid that we have used for genome editing in this study. As shown in Supplementary Figure 9, no appreciable changes in the expression levels of those genes were noted among them. These findings suggest that possible off-target effects induced by the transfection of the empty vector as well as by DSBs do not affect the p53 pathway in our system, although we have used crude transfected cells in each group.

Additionally, we examined the mRNA expression changes in KRAS signaling pathway (KLK7, BMPR1B, and KLK8), under two different conditions: Patu-S parental cells and Patu-S cells transfected with the empty vector alone. No material changes in the expression levels of those genes were observed (Supplementary Figure 10). This experiment confirmed that possible off-target effects induced by the transfection of the empty vector alone do not affect KRAS signaling pathway, although we have used crude transfected cells in each group.

We have used the clones without mutation induced by transfection of the same genome editing plasmid as negative controls (GP2_WT). Using these clones as negative controls, we believe that we were able to see the GP2 mutation-specific gene expression changes in the clones of GP2_V425M, since CRISPR/Cas9-induced possible off-target effects (such as by unexpectedly induced mutations or by the p53 pathway activation⁷) may equally occur in the negative control clones and the GP2_V425M clones. To exclude the effects of clonal variation caused by selection bias, in addition, we have confirmed the expression changes of specific genes using 3 independent clones in each group in Figure 4E. Taken together, the data presented in the original paper (Figure 4E) exclude three possibilities: off-target effects through unexpectedly induced nonspecific mutations, non-specific p53 pathway activation through DSBs alone, and transcriptional changes of nonspecific genes through GP2 mutation-independent clonal variations.

Supplementary Methods

Transient transfection experiments were carried out in approximately 2×10^5 Patu-S cells plated in 6 well plate. Cells were transfected with the empty vector alone, or the sgGP2 cloned plasmid and the repair template using Lipofectamine 3000 (Life

Technologies) according to the manufacturer's instructions. Gene expression was determined by qRT-PCR 72 hr post-transfection. All primer sequences (in sense format) are listed in Table S7.